# Single-Cell Profiling Reveals Immune-Based Mechanisms Underlying Tumor Radiosensitization by a Novel Mn Porphyrin Clinical Candidate, MnTnBuOE-2-PyP^5+^ (BMX-001)

**DOI:** 10.3390/antiox13040477

**Published:** 2024-04-17

**Authors:** Sun Up Noh, Jinyeong Lim, Sung-Won Shin, Yeeun Kim, Woong-Yang Park, Ines Batinic-Haberle, Changhoon Choi, Won Park

**Affiliations:** 1Department of Radiation Oncology, Samsung Medical Center, Seoul 06351, Republic of Korea; chocolatebox919@gmail.com (S.U.N.); camuserik@gmail.com (S.-W.S.); yeeun17.kim@sbri.co.kr (Y.K.); 2Sungkyunkwan University School of Medicine, Seoul 06351, Republic of Korea; 3Department of Health Sciences and Technology, Samsung Advanced Institute for Health Sciences and Technology, Sungkyunkwan University, Seoul 06351, Republic of Korea; aster1217@gmail.com (J.L.); woongyang.park@samsung.com (W.-Y.P.); 4Samsung Genome Institute, Samsung Medical Center, Seoul 06351, Republic of Korea; 5Department of Radiation Oncology, Duke University School of Medicine, Durham, NC 27710, USA; ibatinic@duke.edu

**Keywords:** Mn porphyrin clinical candidate, MnTnBuOE-2-PyP^5+^ (BMX-001; MnBuOE), single-cell RNA sequencing, tumor heterogeneity, tumor microenvironment, radiotherapy, immune-based mechanisms underlying tumor radiosensitization

## Abstract

Manganese porphyrins reportedly exhibit synergic effects when combined with irradiation. However, an in-depth understanding of intratumoral heterogeneity and immune pathways, as affected by Mn porphyrins, remains limited. Here, we explored the mechanisms underlying immunomodulation of a clinical candidate, MnTnBuOE-2-PyP^5+^ (BMX-001, MnBuOE), using single-cell analysis in a murine carcinoma model. Mice bearing 4T1 tumors were divided into four groups: control, MnBuOE, radiotherapy (RT), and combined MnBuOE and radiotherapy (MnBuOE/RT). In epithelial cells, the epithelial–mesenchymal transition, *TNF-α* signaling via *NF-кB*, angiogenesis, and hypoxia-related genes were significantly downregulated in the MnBuOE/RT group compared with the RT group. All subtypes of cancer-associated fibroblasts (CAFs) were clearly reduced in MnBuOE and MnBuOE/RT. Inhibitory receptor–ligand interactions, in which epithelial cells and CAFs interacted with CD8+ T cells, were significantly lower in the MnBuOE/RT group than in the RT group. Trajectory analysis showed that dendritic cells maturation-associated markers were increased in MnBuOE/RT. M1 macrophages were significantly increased in the MnBuOE/RT group compared with the RT group, whereas myeloid-derived suppressor cells were decreased. CellChat analysis showed that the number of cell–cell communications was the lowest in the MnBuOE/RT group. Our study is the first to provide evidence for the combined radiotherapy with a novel Mn porphyrin clinical candidate, BMX-001, from the perspective of each cell type within the tumor microenvironment.

## 1. Introduction

Currently, several anticancer treatment options are available, such as chemotherapy, immunotherapy, hormone therapy, surgery, and radiation. Among those, radiation therapy (RT) may be particularly effective for treating localized or solid cancers. Approximately half of all patients with cancer receive RT as a curative or palliative treatment. Moreover, as an adjuvant, RT is frequently combined with other types of treatment such as chemotherapy and surgery. However, the side effects of RT, which originate from reactive species-driven oxidative stress injury of normal tissue, have further prompted the development of safer and targeted therapies [1,2,3,4]. Radiation paradoxically triggers various changes in the tumor microenvironment (TME) that may lead to the risks of relapse and metastasis.

The differential impact of cationic manganese-substituted pyridylporphyrins on both normal and tumor tissues has been extensively studied [5,6]. These compounds sensitize tumors to radio and chemo treatment and simultaneously protect normal tissue via modulation of their redox status [7]. The effect of Mn porphyrins, commonly known as superoxide dismutase (SOD) mimics, has been studied in various tumors, such as breast, head and neck, prostate, and brain [5,6,8]. The promising data obtained from cellular and animal studies have facilitated the progress of Mn(III) *meso*-tetrakis (*N*-n-butoxyethylpyridinium-2-yl) porphyrin, i.e., MnTnBuOE-2-PyP^5+^ (BMX-001, MnBuOE), into clinic trials. With a good safety/toxicity profile, MnBuOE is presently tested on normal tissue protection, while tumor growth suppression is performed in five phase II clinical trials on patients bearing glioma, head and neck cancer, anal cancer, rectal cancer, and multiple brain metastases [5,6,9]. In addition, a recent glioblastoma study has shown that patients with glioblastoma have improved survival rates when their treatment with irradiation was combined with MnBuOE (BMX-001) [9]. The biocompatible redox properties of Mn porphyrins, their ability to interact with numerous reactive species, their bioavailability within cells and cellular compartments, and the tumor heterogeneity of immunogenic and metabolic pathways necessitate additional studies on the nature of the differential actions of Mn porphyrins within the TME. Previously, we explored the anticancer potency and metabolic pathways affected by an earlier analog, MnTnHex-2-PyP^5+^ [10]. Here, considering the progress of Mn porphyrins into clinical settings, we have explored the complex metabolic pathways that play important roles in the anticancer activities of MnBuOE (BMX-001).

Despite the evidence supporting the role of Mn porphyrins in cancer therapy, little is known about their immunomodulatory effects. Thus far, studies on these compounds have been limited to total RNA sequencing [11]. In a previous study, we assumed that Mn porphyrins could inhibit RT-induced epithelial-to-mesenchymal transition (EMT) in the TME by suppressing pro-survival signaling pathways, the AKT/GSK3β/Snail pathway, and NF-κB activation in a mouse 4T1 cancer cell line in vitro or in vivo [10]. However, our understanding of the molecular mechanisms of Mn porphyrins has been largely limited to the estimation of the average gene expression of tumor cells.

Tumors are intricate ecosystems. The TME is composed of diverse cells, including cancer cells and stromal subsets, whose specific characterization is masked by heterogeneity. Numerous studies have suggested that stromal cells, such as epithelial cells, T cells, macrophages, and fibroblasts, which are highly heterogeneous, are associated with tumors [12,13,14,15,16]. Tumor heterogeneity governs many decisive facets of tumor pathogenesis that are driven by tumor growth, metastasis, and resistance to treatment. Therefore, it is essential to examine the gene expression patterns of individual cells.

Single-cell RNA sequencing (scRNA-seq) enables specific profiling of individual cell populations, thereby enabling unbiased distinguishing of heterogeneous stromal and cancer cells at the resolution of individual cells. Therefore, scRNA-seq techniques have emerged as promising methods for elucidating tumor pathogenesis, revealing the complexities of and differences between the molecular components [17,18,19]. Furthermore, understanding the correlation between cancer and stromal/immune cells in the TME and identifying potential targets could be particularly important for determining the synergistic effect of MnBuOE/RT. In this study, we aimed to explore how Mn porphyrin, MnBuOE (BMX-001), and RT affect the characteristics of tumor and stromal cells in murine mammary carcinoma using scRNA-seq.

## 2. Materials and Methods

### 2.1. Animal Models

For the establishment of the 4T1 tumor model, 6–7-week-old female BALB/c mice were purchased from Orient Bio (Gapyeong, Korea) and cells (1 × 10^5^ cells in 50 μL phosphate-buffered saline) were injected subcutaneously into the right hind leg of each mouse. Tumor volumes were measured every 3 days using calipers and calculated as volume = (width^2^ × length)/2. When the mean tumor volume reached 80–120 mm^3^, the mice were randomly divided into four groups and each experimental group consisted of eight mice: control group: (CN), MnTnBuOE-2-PyP^5+^ group (MnBuOE), radiotherapy group (RT), and group receiving MnBuOE along with radiotherapy (MnBuOE/RT). MnBuOE was injected intraperitoneally (1 mg/kg) twice a week. Two hours after drug administration, irradiation was conducted on the tumor-bearing hind leg over three continuous days at 2 Gy X-ray for a total of 6 Gy. During irradiation, the mice were anesthetized via intraperitoneal injection of 30 mg/kg Zoletil (Virbac, Carros, France) and 10 mg/kg Rompun (Bayer, Leverkusen, Germany), as prescribed by veterinarians. Fifteen days after irradiation, all tumor tissues were isolated and excised. Tumor tissues were prepared with three individuals pooled per group for scRNA-seq. Some tumor tissues were prepared for flow cytometric assay and the other tumor tissues were fixed with 10% formalin and embedded in paraffin for terminal deoxynucleotidyl transferase(TdT)-mediated biotinylated dUTP nick end labeling (TUNEL) analysis. Experimental procedures were repeated at least three times. The study protocol (20220210001) was reviewed and approved by the Institutional Animal Care and Use Committee (IACUC) of the Samsung Medical Center (SMC). SMC is an Association for Assessment and Accreditation of Laboratory Animal Care International accredited facility and abides by the Institute of Laboratory Animal Resources guidelines.

### 2.2. Tissue Dissociation into Single-Cell Suspension

Tumor tissues were dissected from the mice and dissociated into single-cell suspensions via mechanical dissociation combined with enzymatic degradation of the extracellular matrix (ECM), which maintains the structural integrity of tissues. The tumor tissue was enzymatically digested using a Tumor Dissociation Kit (Miltenyi Biotec., Bergisch Gladbach, Germany), and gentleMACS™ Dissociators (Miltenyi Biotec.) were used for mechanical dissociation. After dissociation, a filter was used to remove any remaining larger particles from the single-cell suspension.

### 2.3. Single-Cell RNA Sequencing Data Processing

The single-cell suspensions were washed and loaded onto a Chromium single-cell system (10x Genomics, Pleasanton, CA, USA). The barcoded sequencing libraries were created using the Chromium Single-Cell 5’ Reagent Kits (10x Genomics) according to the manufacturer’s instructions and then sequenced on a Novaseq6000 platform (Illumina, San Diego, CA, USA). The resulting sequencing data were aligned to the mouse reference genome (GRCm38) and processed through the CellRanger 4.0.0 pipeline (10x Genomics). A stringent selection process was imposed to exclude cells that failed to reach the sufficient cell quality threshold. Specifically, cells exhibiting fewer than 500 unique molecular identifier (UMI) counts, fewer than 250 detected genes, more than 30% mitochondrial gene expression, or low cell complexity (l log10GenesPerUMI l ≤ 0.8) were omitted. Ensuring the singularity of cell population and the exclusion of potential doublets, the “DoubletFinder” package (Version 2.0.3) was employed. Consequently, approximately 10% of cells were annotated low quality of cells (4231 of 39,585) and excluded from the subsequent analysis. Single-cell analysis was performed in the Seurat R package (v.4.3). Specifically, the gene expression matrices were normalized and transformed to the log scale. For feature selection, the top 2000 highly variable genes expressed in each sample were chosen.

### 2.4. Cluster Identification and Annotation

For clustering, the variably expressed genes were subjected to a principal component analysis (PCA). The number of principal components selected for the major cluster (PC 20) or subset clusters (PC 19 for Epithelial, PC21 for fibroblasts, PC32 for T/NK cells, PC34 for dendritic cells (DCs), and PC33 for monocytes/macrophage cells) was determined by evaluating the slope of the elbow plot. Both PCA and uniform manifold approximation and projection (UMAP) dimension reduction were performed using the selected PCs. The nearest-neighbor graphs were calculated using the same PC dimensions from the PCA reduction, and clustering was performed with resolutions varying from 0.2 to 0.7 depending on cell types. To determine the cell type for the major cluster or subset clusters, differentially expressed genes (DEGs) were determined using the “findmarker” function in Seurat R package (v.4.3) based on the model-based analysis of single-cell transcriptomics test with a minimal fraction of 25% and a log-transformed fold-change threshold of 0.25 [20,21]. Canonical markers for scRNA-seq data from the relevant literature were used [22]. Additionally, we employed SingleR and scAnnotate for further validation of cell-type annotations with annotated single-cell reference data [23,24,25,26]. To visualize the canonical markers and DEGs, heatmaps, dot plots, and violin plots were generated to show the expression of the markers used for identifying each cell type.

### 2.5. Identifying Cancer Cells in Epithelial Cell Types

For cancer cell prediction, we utilized the CopyKAT package (v.1.1.0) on epithelial cell types [27]. CopyKAT functions were executed with default parameters, without specifying a mouse genome (“mm10”), across individual samples. Cells showing aneuploidy features were identified as potential cancer cells. These predicted cancer cells were then annotated as a distinct cell type for subsequent downstream analysis.

### 2.6. Pathway Enrichment Analysis

To identify biological functions or pathways that were significantly associated with specific cell types or gene sets, we performed a gene set variation analysis (GSVA, v.1.50.0) with the hallmark gene sets from the Molecular Signatures Database (Msigdb, v.7.5.1) using the average gene expression of each cell type or group. Additionally, we conducted a gene set enrichment analysis (GSEA) by ranking the DEGs of each targeted cluster or group according to log-transformed fold change (logFC) and then utilizing this ranked list as input for the fgsea function in the fgsea R package (v.1.28.0) [28].

### 2.7. Trajectory Analysis

Cell lineage analysis in DCs was performed using the monocle v.2 package [29]. We reconstructed the single-cell trajectory by creating a monocle object using the UMI count metrics and the “negbinomial.isze” parameter with default settings. To identify DEGs, we used the differentialGeneTest function to select the top 300 genes with the lowest q-values. Dimensional reduction and cell ordering were conducted using the DDRTree method and the orderCells function, respectively.

### 2.8. Cell–Cell Communication and Receptor–Ligand Interaction Analysis

The cell–cell interactions based on the expression of ligand–receptor pairs in different cell types were inferred using the CellChat R package (v.1.5.0) [30]. We followed the recommended workflow in CellChat and utilized the default settings to identify major signaling interactions and evaluate the coordination of cells and signals for various functions. Briefly, the normalized counts were used as a CellChat object and subjected to the preprocessing functions, including identifyOverExpressedGenes, identifyOverExpressedInteractions, and projectData with the default parameters. The strength of ligand–receptor interactions and the number of interactions were determined using the computeCommunProb, compute-CommunProbPathway, and aggregateNet functions with the default parameters applied in a stepwise manner.

### 2.9. Data and Code Availability

Our data and code used to reproduce the analysis and figures described in this manuscript are available at https://doi.org/10.5061/dryad.b2rbnzspd (17 April 2024). The exact versions of both the R package and analysis code used for this study are also available from zenodo; https://doi.org/10.5281/zenodo.10702330.

### 2.10. TUNEL Staining

Deparaffinized and dehydrated tumor tissue sections were stained by the TUNEL In Situ Cell Death Detection Kit (Roche Applied Science, Mannheim, Germany) according to the manufacturer’s protocol. Briefly, tumor tissue sections were placed in a 3% hydrogen peroxide solution with methanol to block endogenous peroxidase activity and were incubated in 0.1% sodium citrate containing 0.1% Triton X-100 to increase tissue permeability. After rinsing in PBS, 50 µL of the TUNEL reaction mixture (calf thymus TdT and nucleotides) was added to each sample. After incubation at 37 °C in the dark for 60 min, these sections were rinsed with PBS and the apoptotic cells were marked by 3,3′-diaminobenzidine (DAB) through a horseradish peroxidase (HRP) catalysis of biotinylated dUTP-streptavidin-HRP. Images were captured using an Aperio ScanScope AT slide scanner (Leica Biosystems, Inc., Buffalo Grove, IL, USA). Numbers of TUNEL-positive cells were determined with ImageScope software (Version 12.4.6, Leica Biosystems, Inc.).

### 2.11. Flow Cytometric Analysis

Harvested tumors were cut into small pieces and dissociated using a Tumor Dissociation Kit according to the manufacturer’s instructions (Miltenyi Biotec, Auburn, CA, USA). Red blood cells were lysed with BD Pharm Lyse^TM^ lysing buffer (BD Bioscience, San Jose, CA, USA). Cell suspensions were stained with PerCP-Cy5.5-conjugated anti-CD45 antibody, FITC-conjugated anti-mouse CD3 antibody, APC-Cy7-conjugated rat anti-mouse CD4 antibody, V450-conjugated rat anti-mouse CD8 antibody, APC-conjugated anti-mouse CD25 antibody, APC-Cy7-conjugated rat anti-mouse CD45 antibody, PerCP-Cy5.5-conjugated rat anti-mouse CD11b antibody, Alexa Fluor 647-conjugated rat anti-mouse F4/80 antibody, FITC-conjugated anti-CD86 antibody (BD bioscience), or PE-Cy7-conjugated anti-CD206 antibody (eBioscience, San Diego, CA, USA). For intracellular staining, cells were fixed and permeabilized with a fixation/permeabilization buffer (eBioscience) and stained with PE-conjugated rat anti-mouse Foxp3 antibody (BD bioscience). Flow cytometric analysis was performed using a BD FACS Verse flow cytometer (BD bioscience) and FlowJo software version 10.6.1 (BD bioscience).

### 2.12. Statistical Analysis

GraphPad Prism 9.4.1 (GraphPad Software, San Diego, CA, USA) was used for all statistical analyses. Differences among groups were determined by Student’s *t*-test with Bonferroni correction for comparison between two groups or one-way analysis of variance (ANOVA) following Tukey post hoc test. Tumor growth curves were analyzed using a two-way analysis of variance (ANOVA) with Tukey’s correction for multiple comparisons. Statistical significance is presented as * *p* < 0.05, ** *p* < 0.01, *** *p* < 0.001, or **** *p* < 0.0001. The statistical details of each experiment are indicated in the figure legends.

## 3. Results

### 3.1. Effect of MnBuOE Coupled with Irradiation and Major Single-Cell Mapping in the 4T1 Tumor Mouse Model

Inflated tumor volumes around the right hind leg were measured to evaluate the tumor growth trend for the experimental groups (CN, MnBuOE, RT, MnBuOE/RT) after induction of 4T1 tumors (Figure 1A,B). Tumor growth rapidly increased in the CN group after 9 days of tumor randomization (day zero). Compared with the CN group, no significant change in tumor growth was observed in the MnBuOE group. However, significant reduction in tumor growth compared with the CN and MnBuOE groups (*p* < 0.05) was observed in the MnBuOE/RT group. At 15 days, the tumor volume of the MnBuOE/RT group was reduced by approximately 61.2% compared with that of the MnBuOE group.

The scRNA-seq analysis was performed to investigate the heterogeneous transcriptomic responses of tumor, stromal cells, and immune cells within each experimental cohort. Following the application of stringent quality filters to eliminate low-quality cells, the resulting dataset displayed a mean of 2227 transcripts per cell, with an average of 10,362 reads per cell (Appendix A). Following PCA and UMAP visualization, a total of 35,354 single cells sorted from all four groups (CN, MnBuOE, RT, MnBuOE/RT) revealed eight major clusters: monocytes/macrophages (14,177 cells, 40.6% of the total), neutrophils (11,089 cells, 31.8% of the total), T/NK cells (3013 cells, 8.6% of the total), and DCs (consist of plasmacytoid dendritic cell (pDC) and conventional dendritic cell (cDC), 892 cells, 2.5% of the total), epithelial cells (4098 cells, 11.7% of the total), fibroblasts (1390 cells, 3.9% of the total), endothelial cells (135 cells, 0.4% of the total), and myocytes (103 cells, 0.2% of the total) (Figure 1C). The annotation of each cluster was facilitated through broad cell marker genes including *Ptprc* (immune cells), *Lyz2*, *Apoe* (monocytes/macrophages), *Csf3r* (neutrophils), *Cd3e* (T cells), *Cd74* (macrophages and DCs), *Krt7* (epithelial cells), *Col1a1* (fibroblasts), and *Pecam1* (endothelial cells) (Figure 1D and Appendix A). A distinct partitioning of these major cell categories was distinguished into 29,171 immune cells (depicted in blue, 83% of the total) and 5726 non-immune cells (denoted in gray, 16% of the total) (Figure 1E). Of note, the non-immune cell population in CN was most pronounced, constituting 20% of the total cell proportion. In contrast, the MnBuOE/RT group displayed a proportion of non-immune cell clusters, accounting for 15.6% of the total cell (Figure 1F). This reduction in non-immune cell proportions within the treatment group could be attributed to a concurrent reduction in tumors. Notably, among the nonimmune cell types, increased epithelial cells and decreased fibroblast in the MnBuOE treatment group showed clear characteristics (Figure 1G). This intriguing observation showed further comprehensive investigation to elucidate the underlying mechanisms and implications of such a response.

### 3.2. Epithelial Cell Clustering and Subtype Analysis of the 4T1 Tumor Treated with MnBuOE Coupled with Irradiation

To reveal the potential functional subtypes of the overall epithelial cell populations, 4098 epithelial cells were re-clustered and four epithelial related cell clusters were identified. These were identified based on corresponding unique signature genes and assigned to known epithelial cell types (Figure 2A,B and Appendix A). The first cluster, Epi1_Epcam−, was characterized by high expression of the *Twist1* and *Zeb2* genes, which are predominantly associated with mesenchymal cell polarization. In contrast, the second cluster, Epi2_Epcam+, distinguished by specific expression of *Epcam* and *Cdh1* genes, was dominant in epithelial cell polarization. The third cluster, Epi3_cycling, was associated with cell proliferation and was identified by high *Mki67* expression. The remaining cells, which formed the fourth cluster, Epi4_Rps, were excluded from further analyses, owing to their lower quality, and expected as normal cells (Appendix A). As shown in Figure 2D, the distribution patterns of epithelial cell clusters were comparable among experimental groups. The percentage of Epcam− cells in the MnBuOE/RT group (15%) was lower than that in the RT group (35%). Moreover, the Epcam+ fraction in the MnBuOE/RT group (31%) was higher than that in the RT group (20%). The mesenchymal cell marker genes *Twist1* and *Zeb2* were significantly reduced in the MnBuOE/RT group compared with the other groups, and the epithelial cell marker gene *Cdh1* was increased in the MnBuOE/RT group compared with the RT group (*p* < 0.0001, Figure 2E). Additionally, the expression of the *Vegfa* gene associated with angiogenesis and the *Hif1a* gene associated with hypoxia were significantly lower in the MnBuOE/RT group than in the other groups (*p* < 0.001). The *Tgfb1* gene related to the inhibition of T-cell activation was significantly downregulated in the MnBuOE/RT group compared with the RT group (*p* < 0.0001).

GSVA was performed to determine the biological functions of each epithelial subtype in tumorigenicity and progression. The Epcam− and Epcam+ clusters had significantly enriched GSVA scores for pathways such as EMT, *TNF-alpha* signaling via *NF-κB*, inflammatory response, angiogenesis, and hypoxia (Figure 2C). In contrast, the cycling cluster showed high expression of genes involved in cell cycle pathways such as G2M checkpoint, E2F target, and DNA repair. As shown in Figure 2F, we filtered a total of 90 DEGs using the criteria of |log2-fold change| ≥ 0.5 and *p* < 0.05 between the MnBuOE/RT and RT groups. These DEGs were divided into two groups, containing 34 upregulated and 56 downregulated genes. We annotated these DEGs into hallmark gene pathways, using Msigdb, and color-coded the relevant pathways (EMT, *TNF-alpha* signaling via *NF-κB*, inflammatory response, angiogenesis, and hypoxia) in a volcano plot. Additionally, the DEGs were subjected to GSEA to investigate the key pathways and core genes between the RT and MnBuOE/RT groups. Figure 2G shows the GSEA enrichment plot of five hallmark pathways in the MnBuOE/RT group compared with the RT group, namely, EMT (*Serpine2, Mgp, Vim, Vegfa* and others), *TNF-alpha* signaling via *NF-κB* (*Ccl2, Ptsg2, Cebpb, Areg, Btg3,* and others), inflammatory response (*Ifngr2, Axl, Selenos, Mmp14, Cd82,* and others), angiogenesis (*App, Slco2a1, Vegfa, Spp1,* and *Lrpap1*), and hypoxia (*Ppp1r15a, Prdx5, Aldoc, Phha2, Ldha,* and others). We observed that the biological pathways relevant to EMT, *TNF-alpha* signaling via *NF-κB*, inflammatory responses, angiogenesis, and hypoxia were significantly downregulated in the MnBuOE/RT group compared with the RT group (*p* < 0.05). In particular, apoptosis-related gene analysis using Msigdb did not show discrimination between experimental groups, but TUNEL staining showed significantly higher TUNEL-positive cells in the MnBuOE/RT group compared with the rest of the groups, indicating an increased apoptosis (*p* < 0.001, Figure 2H,I).

### 3.3. Fibroblasts Clustering and Subtype Analysis of the 4T1 Tumor Treated with MnBuOE Coupled with Irradiation

The 1390 fibroblasts were clustered in three separate subsets (Appendix A) corresponding to inflammatory, myofibroblastic, and cycling cancer-associated fibroblasts (CAFs). We observed distinct phenotypic differences that enabled the characterization of their functions in greater detail (Appendix A). For instance, chemokine markers involved in angiogenesis or inflammation such as *Cxcl1, Cxcl2,* and *Cxcl12* were mainly upregulated in inflammatory CAFs, whereas myogenic markers such as *Tagln, Acta2,* and *Mmp9* showed the highest expression in myofibroblastic CAFs. Expression of the cell proliferative marker *Mki67* was increased in cycling CAFs. Remarkably, most of the CAFs, including myofibroblastic and inflammatory CAFs, substantially declined in the MnBuOE treatment groups (MnBuOE, MnBuOE/RT) compared with the CN and RT groups (Appendix A). We confirmed that the number of myofibroblastic CAFs decreased from 39 to 6 and that of inflammatory CAFs decreased from 143 to 7 in the MnBuOE/RT group compared with those in the RT group. These results showed that MnBuOE treatment resulted in CAF removal in the 4T1 tumor model.

### 3.4. T Cells Clustering and Subtype Analysis of the 4T1 Tumor Treated with MnBuOE Coupled with Irradiation

The 3013 T/NK cells were re-clustered into five clusters designated as natural killer (NK) T cells, naive T cells, CD8+ effector memory T cells (Cd8_Tem), CD4+ regulatory T cells (Cd4_Treg), and cycling T cells by UMAP plotting (Figure 3A). As shown in Figure 3B and Appendix A, the functional description in the T cell compartment was determined by specific gene expression of each cluster: NK (*Gzma, Tyrobp, Fcer1g*), T naïve (*Tcf7, Ccr7*), Cd8_Tem (*Cd3d, Cd3e, Cd8a, Gzmk, Lag3, Nkg7*), Cd4_Treg (*Cd4, Foxp3, Tnfrsf9, Ctla4*), and cycling T (*Pclaf, Mki67, Stmn1*). Subsequently, we analyzed the distribution patterns of T cell clusters among all groups (Figure 3C). Based on the distribution of T naïve and cycling T cells, the proportion of the total T cell population indicated that the lowest manifestation occurred in the MnBuOE/RT group. However, the ratio of Cd8_Tem was increased in the MnBuOE/RT group compared with the other groups. The ratio of Cd4_Treg was decreased in the MnBuOE/RT group compared with the RT group. When assessing tumor-infiltrating exhausted CD8+ T cells with exhaustion unique gene markers, such as *Lag3, Tigit, Havcr2, Ctla4,* and *Pdcd1,* we observed that the *Lag3* and *Tigit* levels in the MnBuOE/RT group were significantly lower than those in the other groups (*p* < 0.05, Figure 3D). T cell dysfunction scores were significantly lower in the MnBuOE/RT group than in the other groups, consistent with our data on lower levels of Treg cells and exhausted CD8+ T cells (*p* < 0.01, Figure 3E). These results provided evidence for the differential distribution of T cell clusters, indicating an improvement in the TME of the MnBuOE/RT group. Considering the effect of Cd8_Tem in distinguishing the TME between the RT and MnBuOE/RT groups, we investigated receptor–ligand interactions across fibroblastic CAFs or epithelial cell clusters, including Cd8_Tem. The heatmap plot function was used to analyze the specific receptor–ligand interactions (rows) between two different cell types (columns) (Figure 3F). Inhibitory receptor–ligand pairs, such as T cell immunoglobulin and immunoreceptor tyrosine-based inhibitory motif domain (*Tigit*)-poliovirus receptor (*Pvr*), *Tigit-Nectin2* and *Tigit-Nectin3* between Cd8_Tem, and three clusters of fibroblastic CAFs, were rarely observed in the MnBuOE/RT group compared with the RT group (*p* < 0.01). Similarly, *Tigit* binding affinities for *Pvr* and *Nectin2* between Cd8_Tem and epithelial cell clusters (Epcam−, Epcam+, and cycling) were lower in the MnBuOE/RT group than in the RT group (*p* < 0.01). Furthermore, costimulatory interactions between Cd8_Tem and epithelial cell clusters were significantly increased in the MnBuOE/RT group compared with the RT group (*p* < 0.01), from which specific receptor–ligand complexes, *Cd226-Pvr* and *Cd226-Nectin2*, were identified. Our results suggest that MnBuOE treatment combined with irradiation in tumors can reduce *Tigit*-mediated inhibition of CD8 + T cells while increasing *Cd226*-mediated T cell stimulation. Additionally, we conducted a flow cytometry of T cells in the experimental groups. The total T cell population and CD8 effective T cells were more significantly increased in the MnBuOE/RT group than in the CN group and the RT group (*p* < 0.05, Figure 4A). While Foxp3+ regulatory T cells (Tregs) tended to decrease in the MnBuOE/RT group compared with the other groups, the ratio of CD8 T cells to Tregs increased in the MnBuOE/RT group compared with the rest of the groups (*p* < 0.05, Figure 4B).

### 3.5. DC Clustering and Pseudotime Trajectory Analysis of the 4T1 Tumor Treated with MnBuOE Coupled with Irradiation

To investigate the differentiation trajectory of DC clusters, the monocle v2 package of R software was used for the trajectory analysis. We detected four sorted DC subtypes: DC_Cd40+, cDC1_Cd103+, cDC2_Cd11b+, and pDC_Siglech+ (Figure 5A and Appendix A). As shown in Figure 5B, most cells from each cluster were collected based on gene signatures and the three clusters were formed via a relative process in pseudotime. The process started with the first cluster, cDC1, as they exhibited the highest level of Cd103, followed by the second cluster, cDC2, with the highest level of Cd11b, and ended with the third cluster, DC_Cd40+. The cluster of pDC equivalent to SiglecH was excluded due to its low abundance. The violin plots showed that the DC_Cd40+ cluster had high levels of various surface protein markers, such as Cd40 and Cd86, which are important for DC maturation and T cell co-stimulation, and chemokine receptor Ccr7, which is involved in the migration of DCs to lymph nodes (Figure 5C). Moreover, we observed that the proportion of the DC_Cd40+ cluster increased twofold in the MnBuOE/RT group compared with the other groups, which is consistent with the high expression of Cd40, Cd80, Cd86, and Ccr7 in the MnBuOE/RT group (Figure 5D,E). These results indicated that MnBuOE treatment combined with irradiation facilitates DC maturation under tumor circumstances.

### 3.6. Macrophage Clustering and Phenotype Analysis of the 4T1 Tumor Treated with MnBuOE Coupled with Irradiation

The 14,177 macrophages were clustered in seven separate subsets (Figure 6A). One cluster corresponded to myeloid-derived suppressor cells (MDSCs) (*Ly6C+, Itgam+, Cxcl3+, Il1a+, Il1b+*), two corresponded to M0-like macrophages (MHC II+ type and Cx3cr1+ type), one corresponded to M1-like tumor-associated macrophage (TAM) (*IL1b+, Cxcl3+, Il1a+, Mmp12+*), two corresponded to M2-like TAMs (Arg1+ TAM1 type and Cd206+IL10+ TAM2 type), and the final cluster corresponded to cycling macrophages (*Pclaf+, Tubb5+*) (Figure 6B and Appendix A). Based on the distinction in gene signature scores between M1-like TAM, M2-like TAM, and MDSC subgroups, we examined the TAM polarization and estimated the change in the antitumor inflammatory reactions for all experimental groups. We observed a significant increase in the M1/M2 macrophage phenotype ratio and a significant decrease in the MDSC subgroup in the MnBuOE/RT group compared with the other groups (Figure 6C,D). The flow cytometry results of CD86/206 for the macrophage phenotype in the experimental groups also showed that the ratio of the M1/M2 phenotype increased in the MnBuOE/RT group compared with the other groups (*p* < 0.05, Figure 6E,F). Moreover, macrophages in the MnBuOE/RT group showed significantly higher expression of inflammatory cytokines (*IL-1a, IL-1b*) and C-C motif chemokine ligands 3/4 (*Ccl3, Ccl4*) in M1-phenotype macrophages than those in the RT group (*p* < 0.0001, Figure 6G). These results suggest that MnBuOE treatment combined with irradiation can lead to decreased MDSCs accompanied by an M1 macrophage bias, which results in an anti-tumorigenic TME in tumors.

### 3.7. Intercellular Communications within the TME Regulated by MnBuOE Coupled with Irradiation in the 4T1 Tumor Mice Model

To investigate potential interactions across different cell types in the TME, cell-to-cell communication analysis was performed using the CellChat database, a publicly available repository of curated receptors and ligands and their interactions. As shown in Figure 7A, the heatmap plot illustrated potential interplay with combinations across all cell subtypes between MnBuOE/RT and RT groups. The color of each square indicated relative values, based on the absolute count calculated for the intercellular interaction score of MnBuOE/RT group minus the RT group. The results showed that the number of interactions from epithelial cells, DCs, and macrophages to T cells was increased in the MnBuOE/RT group compared with the RT group. The interaction within the TME (from fibroblasts to endothelial cells) was decreased in the MnBuOE/RT group compared with the RT group. Subsequently, we investigated the number of inferred interactions, including cell–cell contact, ECM–receptor, and secreted signaling, for all experimental groups (Figure 7B). The extent of interaction for each intercellular link was lowest in the MnBuOE/RT group. These results imply that MnBuOE treatment combined with irradiation may lead to T cell communication toward DC maturation, EMT suppression, and M1 phenotypic polarization in the TME, followed by suppression of the formation of CAFs and tumor endothelial cells.

## 4. Discussion

RT remains the standard-of-care for cancer therapy; however, radiation-induced damage to normal tissues limits its effectiveness in tumor therapy. The number of cells, animals, and clinical studies have validated the signal pathways and gene expressions regulated by Mn porphyrin-based drugs, and we have also reported related genes when Mn porphyrin and radiation combination treatment induces cancer cell death, immune activity, and metastasis inhibition through tumor model experiments. These data support the importance of single-cell unit analysis to identify new therapeutic targets for heterogeneous tumors. Here, we report for the first time a comprehensive characterization of the TME following the treatment with Mn porphyrin clinical candidate, MnBuOE (BMX-001), combined with irradiation using scRNA-seq, focusing on the various multifaceted tumor subpopulations. Progress in scRNA-seq technology has enabled the compositional analysis of the immune system at single-cell levels and permitted the identification and sub-clustering of major cell subsets of the TME, exploration of cell type-, molecular pathway- and etiology-specific gene signatures, and prediction of putative cell–cell interactions [31,32,33]. Our analysis identified eight distinct cell populations with UMAP clustering of tumor tissues. Those were mapped into four non-immune types of epithelial cells, fibroblasts, endothelial cells, and myocytes and four immune clusters of macrophages, neutrophils, T cells, and DCs.

Among the epithelial cell subtypes, the unique functions of Epi1_Epcam− cells and Epi2_Epcam+ cells were strongly associated with EMT, which can invade surrounding tissues and travel through the peripheral circulation. The distribution of these epithelial subtypes enables the understanding of the metastasis hypothesis in epithelial cells. The GSEA of epithelial cells confirms that enrichment pathway scores associated with EMT, *TNF-alpha* signaling via *NF-κB*, angiogenesis, and hypoxia were significantly decreased in the MnBuOE/RT group compared with the RT group, which is consistent with our previous report [10]. Furthermore, we divided fibroblasts into myofibroblastic, inflammatory, and cycling CAFs. CAFs have been reported as a key component of the TME [16] and are a strong source of chemokine CXCL12 and rich in alpha smooth muscle actin-positive cells, which promote tumor growth and angiogenesis and remodel the ECM [34,35,36]. Additionally, CAFs inhibit the function of CD8+ T cells, promote Treg recruitment, and suppress their tumor cell killing abilities by reducing T cell infiltration into the tumor, thus impeding T cell trafficking within the TME and inhibiting cytotoxic activity [37]. In this study, we observed that treatment with MnBuOE reduces CAFs, which may substantially contribute to the prevention of cancer metastasis.

Subsequently, we investigated five major T cell clusters, including NKT, T naive, CD8+ effector memory T cells, CD4+ Tregs, and the remaining T cells to reveal the intrinsic structure and potential functional subtypes of the overall T cell populations. T cells within the TME are prone to either dysfunction or exhaustion, thus preventing CD8+ T cells from eliciting sufficient T cell-mediated killing of tumor cells [14,38]. Our data demonstrate that the proportion of CD8+ effector memory T cells increased, whereas CD4+ Tregs decreased when mice were treated with MnBuOE/RT compared with treatment with RT alone. CD8+ T cells, in particular, are important targets in cancer immunotherapy, making them the focus of numerous single-cell studies. Building on these studies, we recapitulated the heterogeneity of CD8+ T cells according to the cytotoxic, dysfunctional, and naïve-like cell states. The expression of CD8+ T cell exhaustion markers, such as *Lag3* and *Tigit*, was significantly lower in the MnBuOE/RT group than in the RT group, leading to alleviation of T cell dysfunction and restoration of T cell infiltration.

Additionally, intimate cell–cell communications across CAFs or epithelial cell clusters, including CD8+ T cells, were analyzed within the TME. We observed a significant decrease in coinhibitory interactions, such as those among *Tigit-Pvr*, *Tigit-Nectin2*, and *Tigit-Nectin3* in the MnBuOE/RT group compared with the RT group, while a significant increase in costimulatory interactions, such as those between *CD226-Pvr* and *CD226-Nectin2* in the MnBuOE/RT group compared with the RT group. The correlation of high *Tigit* expression with a poor clinical outcome is consistent with the view that one of the functions of *Tigit* is the formation of an immunosuppressive TME [39,40,41]. While Tigit functions as an inhibitory receptor, *CD226* has been known to play important roles in T cell priming and activation. When focusing on the CD8+ T cells in these studies, we demonstrated that MnBuOE treatment combined with irradiation may control the *Tigit/CD226* imbalance by suppressing the exhaustion of CD8+ T cells within the TME.

DCs are essential for T cell-mediated cancer immunity [42]. In particular, the distinguishing directivity of cDCs to stimulate T cells leads to the maturation of DCs and the expression of CD40, CD80, and CD86. We identified the differentiation trajectory of three clusters of DCs, which were formed via a relative process in pseudotime. The differentiation trajectory begins with the CD103+ dominant cDC1 subtype and proceeds with the CD11B+ dominant cDC2 cluster and CD40+ dominant mature DC cluster. We observed a significant increase in the proportion of CD40+ dominant mature DCs in the MnBuOE/RT group compared with that in the other groups. Our results, along with those of previous reports, suggest that the efficacy of MnBuOE/RT treatment leads to the DC maturation and co-stimulation between DCs and CD8+ T cells, thereby resulting in tumor regression.

Accumulating evidence suggests that TAMs are a heterogeneous group of cells with multiple mechanisms involved in promoting tumor progression [43,44,45]. MDSCs are another heterogeneous population of cells that expand during cancer progression, which can also suppress T cell responses. In our scRNA-seq analysis data, macrophages were largely separated into unpolarized M0-like macrophages and polarized M1 and M2 macrophages, including MDSCs. Studies indicate that the M1-like macrophages are pro-inflammatory and release various cytotoxic molecules that are crucial to suppressing tumorigenesis. Conversely, the predominance of M2-like macrophages causes tumor progression. We observed a clear increase in the M1/M2 ratio and a significant decrease in MDSCs in the MnBuOE/RT group compared with the other groups. We further confirmed that both inflammatory cytokines, IL-1a and IL-1b, and chemokines CCL3 and CCL4 inducing M1-polarization, which were reduced by irradiation, were significantly increased following the MnBuOE/RT treatment. Our results, along with those of previous reports [8], suggest that MnBuOE/RT can reduce levels of M2-polarized macrophages while inhibiting MDSCs induced by irradiation of tissues.

Analysis of cell–cell interactions based on the expression of ligand–receptor pairs in different cell types can aid an understanding how intimately major cell types interact. Our results showed that the number of T cell trafficking of DCs, macrophages, and epithelial cells was higher in the MnBuOE/RT group than in the RT group, whereas the number of T-cell exchanges of fibroblasts and endothelial cells was low. In particular, the MnBuOE/RT group had fewer affected ECM receptors or activated intracellular signaling pathways than the other groups.

We attempted to cluster the neutrophils, which account for the second largest proportion of immune cells, and annotate them with proper subtypes. Despite being able to divide the neutrophil population into five clusters, we struggled to assign them with specific biological functions (Appendix A). Due to the lack of specific cell-type markers and functional annotations in neutrophils that scRNA-seq technology can address, there were still limitations in describing specific biological functions of neutrophil subsets such as tumor-associated neutrophils (TANs) in this study. We also tried to distinguish between cancer cells and normal cells using CopyKAT program. For example, out of the 4098 epithelial cells analyzed, 3327 (81%) were identified as aneuploid (tumor cells), 748 (18%) as diploid (normal cells), and 23 (0.5%) remained undefined (Appendix A). Notably, only 227 diploid cells were founded in the “Epi_Epcam−” and “Epi_Epcam+” cell types (2279 cells), which primarily focusing on epithelial cell analysis, confirming that the analysis results of all epithelial cells were similar to those of epithelial cells, except diploid.

In our study, we acknowledge the limitations associated with parameter sensitivity in the clustering process, as implemented by Seurat. Parameter sensitivity can cause variability in the clustering results, but we tried to mitigate this by reproducing the annotations using multiple annotation tools, which allowed us to cross-validate our findings and provide a more robust characterization of the data (Appendix A).

Identifying subclusters of certain cell types using scRNA-seq provided us with novel knowledge, but adequate validation using clinical samples is necessary [46]. To the best of our knowledge, studies on combined treatment of Mn porphyrin and radiation therapy are limited to bulk sequencing in preclinical experiments. The randomized controlled trials and single-cell analysis using clinical samples have not been reported yet. We have presented single-cell data on the effects of Mn porphyrin and radiation therapy in mouse experiments but acknowledge the limitations in validating them. Full validation of the effectiveness of Mn porphyrin and radiation therapy will be an essential next step.

This study aimed to corroborate previous findings that Mn porphyrins can eliminate cancer cells and protect normal cells at the single-cell level through scRNA analysis of cells obtained from tumor tissues. Thus, our data have demonstrated that the treatment of cancer with a combination of MnBuOE and RT could increase CD8 + T cytotoxicity through DC maturation and inflammatory-like macrophages and directly kill tumor cells by lowering the interaction of exhausted CD8+ T cells with epithelial cells and fibroblasts. It could also decrease levels of CAFs and prevent epithelial cells from progressing to EMT, angiogenesis, and inflammation, resulting in the protection against the damage to the surrounding normal cells (Figure 8).

Taken together, our study provides another perception of the anticancer effects of MnBuOE/RT using the database of genes related to the signaling pathways of each cell type within the TME.

## 5. Conclusions

Mn porphyrins, commonly known as SOD mimics, have been reported to increase radiation treatment efficiency by protecting normal tissues while carrying out the radiosensitization of tumors; promising clinical results were demonstrated in various cancers. However, it is necessary to understand the mechanism of action of Mn porphyrins with regards to the heterogeneity of various cells in the tumor microenvironment. For the first time, we explored here the mechanisms underlying the immunomodulation by MnTnBuOE-2-PyP^5+^ (BMX-001), using single-cell analysis in a murine mammary carcinoma model. We demonstrated that MnTnBuOE-2-PyP^5+^, when combined irradiation, was able to kill tumor cells and prevent metastasis by increasing CD8+ T cells through dendritic cell activation and increasing M1 macrophages, while reducing the cancer-associated fibroblasts and inhibiting the angiogenesis and inflammatory responses of epithelial cells. This study reveals the synergistic anticancer effect of BMX-001/radiotherapy at a single-cell level for various cells that comprise tumor heterogeneity, therefore presenting additional insight into the tumor microenvironment.

## Figures and Tables

**Figure 1 antioxidants-13-00477-f001:**
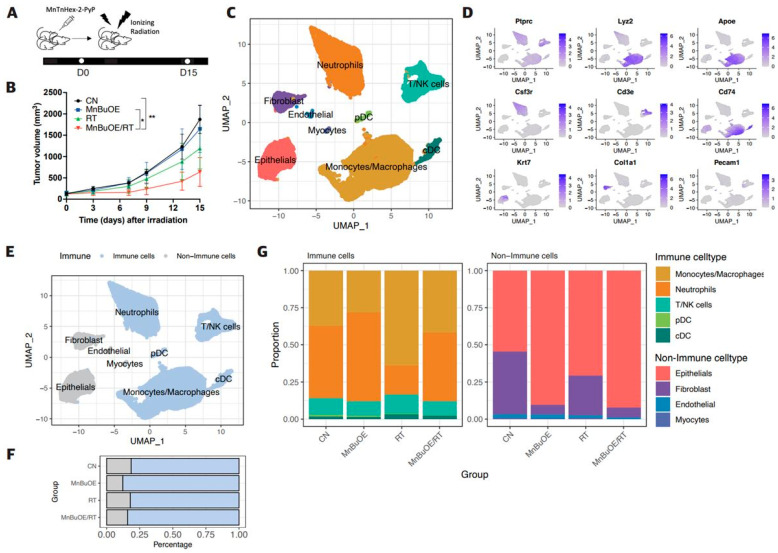
Effect of MnBuOE coupled with irradiation on major single-cell clustering in the 4T1 tumor mice model. (**A**,**B**) Each experimental group consisted of eight mice. The tumor volumes were measured once every 3 days. MnBuOE combined with radiation therapy increased tumor growth delay compared with MnBuOE monotherapy. Tumor-bearing tissues were prepared with three individuals pooled per group for scRNA-seq 15 days after MnBuOE treatment and irradiation. * *p* < 0.05, ** *p* < 0.01. (**C**) After PCA and UMAP analysis of UMI levels in a total of 35,354 single cells isolated from all four groups (CN, MnBuOE, RT, MnBuOE/RT), the hierarchical clustering distinguished eight major clusters: monocytes/macrophages, neutrophils, T/NK cells, DCs, epithelial cells, fibroblasts, endothelial cells, and myocytes. (**D**) According to the expression of specific marker genes for various cell types, immune cell types (*Ptprc*), monocytes/macrophages (*Lyz2, Apoe*), neutrophils (*Csf3r*), T/NK cells (*Cd3e*), DCs (*Cd74*), epithelial cells (*Krt7*), fibroblasts (*Col1a1*), and endothelial cells (*Pecam1*) were identified. (**E**–**G**) The eight main cell subtypes were divided into immune cells and non-immune cells. Blue dots represent immune cells (monocytes/macrophages, neutrophils, T/NK cells, and DCs), whereas grey dots represent non-immune cells (epithelial cells, fibroblasts, endothelial cells, and myocytes). MnBuOE treatment increased the proportion of immune cells compared with the CN group. scRNA-seq, single-cell RNA sequencing; MnBuOE, MnTnBuOE-2-PyP^5+^; PCA, principal component analysis; UMAP, uniform manifold approximation and projection; UMI, unique molecular identifier; DCs, dendritic cells.

**Figure 2 antioxidants-13-00477-f002:**
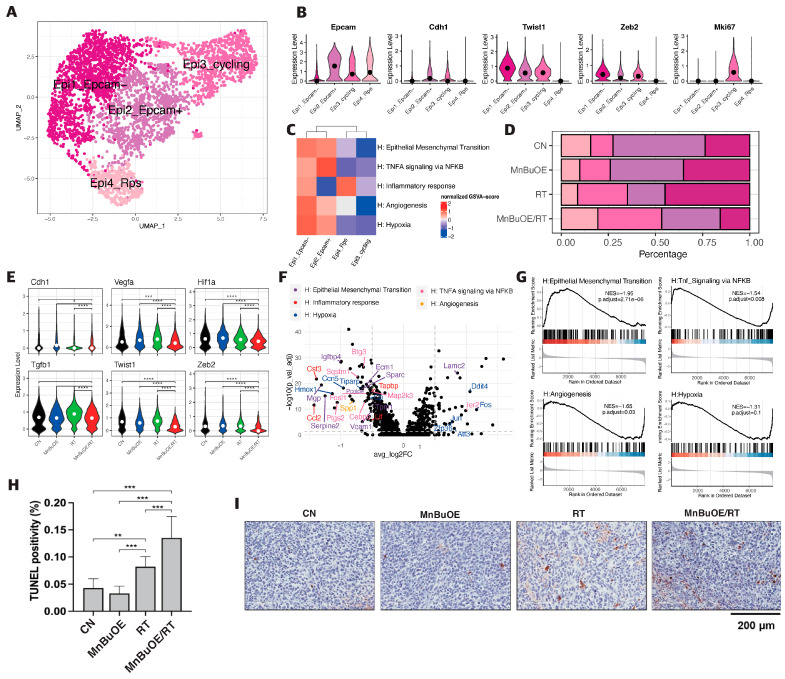
Effect of MnBuOE treatment and irradiation on cancer-associated pathways in epithelial cell subtypes of 4T1 tumor mice. (**A**) UMAP plot of epithelial cells indicates four subtypes: Epicam-(mesenchymal bias), Epicam+ (epithelial bias), cycling (proliferation), and Rps (non-function) types (n = 4098). (**B**) Violin plots show that expression of specific marker genes, including epithelial bias (*Epcam, Cdh1*), mesenchymal bias (*Twist1, Zeb2*), and cell proliferation (*Mki67*), differed between the different epithelial cell subtypes. (**C**) Heatmap of GSVA scores determined using Hallmark GeneSet (EMT, *TNF-alpha* signaling via *NF-κB*, inflammatory response, angiogenesis, and hypoxia) showed increased involvement in the Epicam- and Epicam+ subtypes for each experimental group. (**D**) All experimental groups included the fractions of four clusters of epithelial cell subtypes. (**E**) Violin plots indicated differential expression of *Cdh1*, *vegfa*, *Hif1*, *Tgfb1*, *Twist1*, and *Zeb2* between the different experimental groups. * *p* < 0.05, *** *p* < 0.001, **** *p* < 0.0001. (**F**) Volcano plot of DEGs in the MnBuOE/RT (MnBuOE and irradiation) versus RT (irradiation only) groups. The dot color represents the upregulated or downregulated DEGs, whereas the text color represents the related Hallmark pathway. (**G**) GSEA enrichment plots in relevant Hallmark gene sets in the MnBuOE/RT versus RT groups showing the NES score and adjusted *p*-values. The positions of gene set members on the rank-ordered list indicate the level of enrichment of the genes within the gene set, with a color gradient from red to blue indicating the upregulated to downregulated trend. These results indicate that MnBuOE combined with radiation therapy markedly declined carcinogenesis pathways such as EMT, *TNF-alpha* signaling via *NF-κB*, inflammatory response, angiogenesis, and hypoxia compared with radiation therapy only. (**H**,**I**) TUNEL analysis and DAB staining showed increased apoptosis in tumor tissue sections of the MnBuOE/RT groups compared with the other groups. ** *p* < 0.01, *** *p* < 0.001. MnBuOE, MnTnBuOE-2-PyP^5+^; UMAP, uniform manifold approximation and projection; GSVA, gene set variation analysis; EMT, epithelial-to-mesenchymal transition; DEGs, differentially expressed genes; MnBuOE/RT, MnTnBuOE-2-PyP^5+^/radiation therapy; GSEA, gene set enrichment analysis; NES, normalized enrichment score; TUNEL, terminal deoxynucleotidyl transferase-mediated biotinylated dUTP nick end labeling; DAB, 3,3′-diaminobenzidine.

**Figure 3 antioxidants-13-00477-f003:**
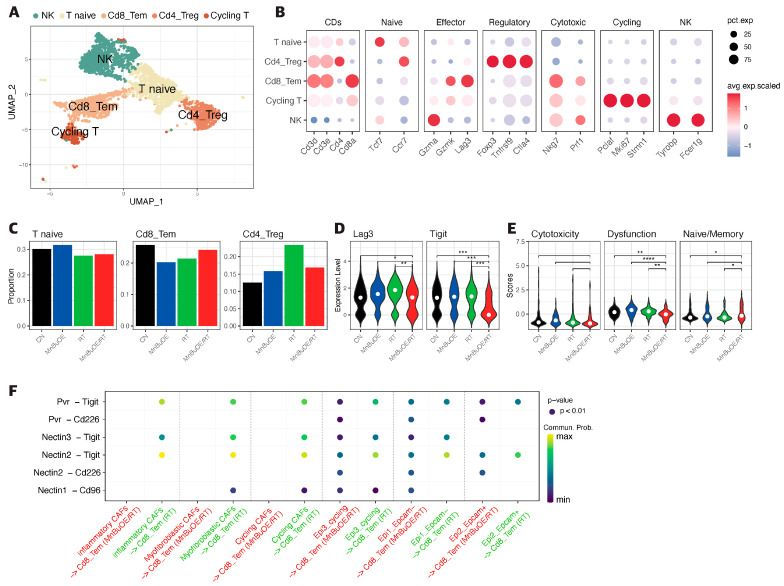
Decreased *Tigit* expression and cell–cell interaction in CD8 effector T cells following treatment with MnBuOE and irradiation. (**A**) UMAP plot of T cells indicates the formation of five main clusters shown in different colors (n = 3013). (**B**) Dot plot of canonical T cell markers in each subtype, i.e., NK (*Gzma, Tyrobp, Fcer1g*), T naïve (*Tcf7, Ccr7*), Cd8_Tem (*Cd3d, Cd3e, Cd8a, Gzmk, Lag3, Nkg7*), Cd4_Treg (*Cd4, Foxp3, Tnfrsf9, Ctla4*), and cycling T (*Pclaf, Mki67, Stmn1*). Circle size represents the percentage of expressed cells in the subtypes, and color indicates the normalized expression. (**C**) Proportion of T cell subtypes in each experimental group. (**D**) Differential expression of inhibitory receptors (*Lag3* and *Tigit*) in Cd8_Tem among the different experimental groups. The exhausted CD8 + T cell markers, *Lag3* and *Tigit,* were significantly downregulated in the MnBuOE/RT (MnBuOE and irradiation) group compared with the RT (irradiation only) group. * *p* < 0.05, ** *p* < 0.01, *** *p* < 0.001. (**E**) Violin plot showing the expression of genes related to cytotoxicity (*Gzma, Prf1*), dysfunction (*Pdcd1, Lag3, Tigit, Havcr2, Ctla4*), and naïve/memory (*Tcf, Ccr7, Il7r*) scores in Cd8_Tem among the different experimental groups. * *p* < 0.05, ** *p* < 0.01, **** *p* < 0.0001. (**F**) Significant inhibitory receptor–ligand pairs (*Tigit* and *Nectin/Pvr*) or stimulatory receptor–ligand pairs (*Cd226* and *Nectin/Pvr*) sending signals from CAFs and epithelial cells to Cd8_TemCircle size represents the levels of significance, and color shows the probability of communication in each pair. MnBuOE, MnTnBuOE-2-PyP^5+^; UMAP, uniform manifold approximation and projection; NK, natural killer; Cd8_Tem, CD8+ effector memory T cells; Cd4_Treg, CD4+ regulatory T cells; Lag3, lymphocyte activation gene 3; *Tigit,* T cell immunoglobulin and immunoreceptor tyrosine-based inhibitory motif domain; MnBuOE/RT, MnTnBuOE-2-PyP^5+^/radiation therapy; *Pvr,* poliovirus receptor.

**Figure 4 antioxidants-13-00477-f004:**
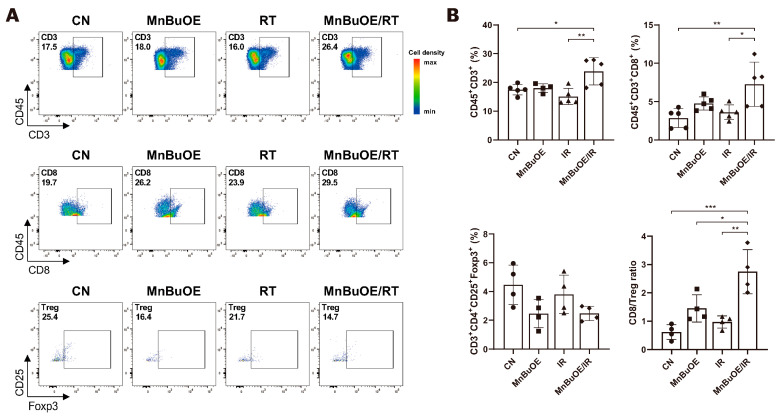
Flow cytometric analysis of the effects of MnBuOE and irradiation on T cell populations. (**A**) Flow cytometric analysis of total T cell population, CD8 + T cells, and Treg cells infiltrated into tumors in each experimental group. Representative density plots are shown. (**B**) The proportion of the total T cell population, CD8 + T cells, and Treg cells is calculated. These results imply that the combination of MnBuOE and irradiation can lead to increased CD8 + T cells and decreased Treg cells while raising total T cell population. * *p* < 0.05, ** *p* < 0.01, *** *p* < 0.001. MnBuOE, MnTnBuOE-2-PyP^5+^; Treg cells, regulatory T cells.

**Figure 5 antioxidants-13-00477-f005:**
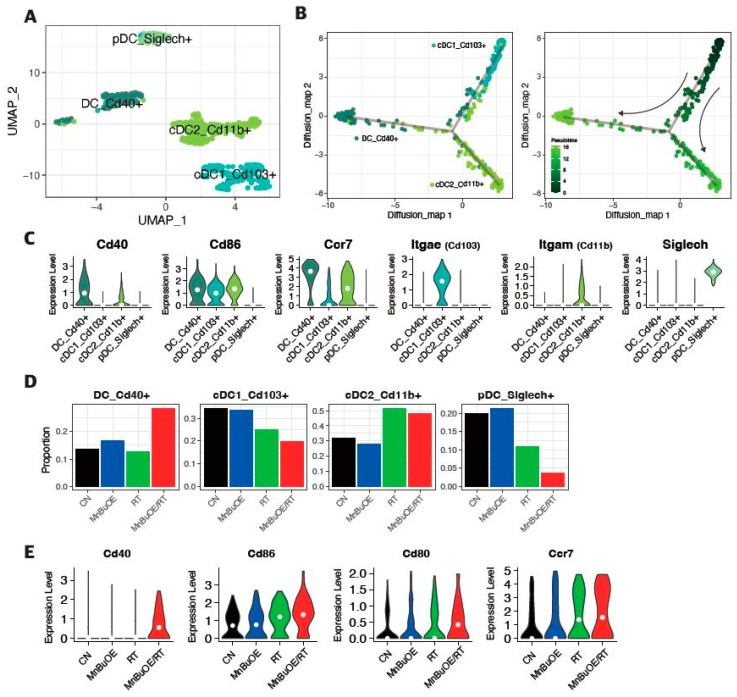
Effect of MnBuOE and irradiation on dendritic cell activation. (**A**) UMAP plot of DCs. DCs clustered in four subsets: DC_Cd40+, cDC1_Cd103+, cDC2_Cd11b+, and pDC_Siglech+. (**B**) Inferred trajectory plot using Monocle2, which shows the color-coded subtypes of DC (**left**) and their corresponding pseudotime (**right**). (**C**) Violin plot showing the expression of canonical markers in each DC subtype. The cDC2_Cd11b+ dominated *Itgam*, the cDC1_Cd103+ dominated *Itgae*, the DC_Cd40+ dominated *Cd86, Cd40, Ccr7,* and the pDC_Siglech+ dominated *Siglech.* (**D**) Proportion of DC subtypes in each experimental group. (**E**) Differential expression of co-stimulator genes (*Cd40, Cd80,* and *Cd86*) and DC markers (*Ccr7*) among the different experimental groups. Mature DCs were identified based on the expression of marker genes such as *Cd40, Cd80, Cd86,* and *Ccr7*. MnBuOE combined with radiation therapy triggered mature DC activation compared with the other groups. MnBuOE, MnTnBuOE-2-PyP^5+^; UMAP, uniform manifold approximation and projection; DCs, dendritic cells; cDC, conventional dendritic cell; pDC, plasmacytoid dendritic cell.

**Figure 6 antioxidants-13-00477-f006:**
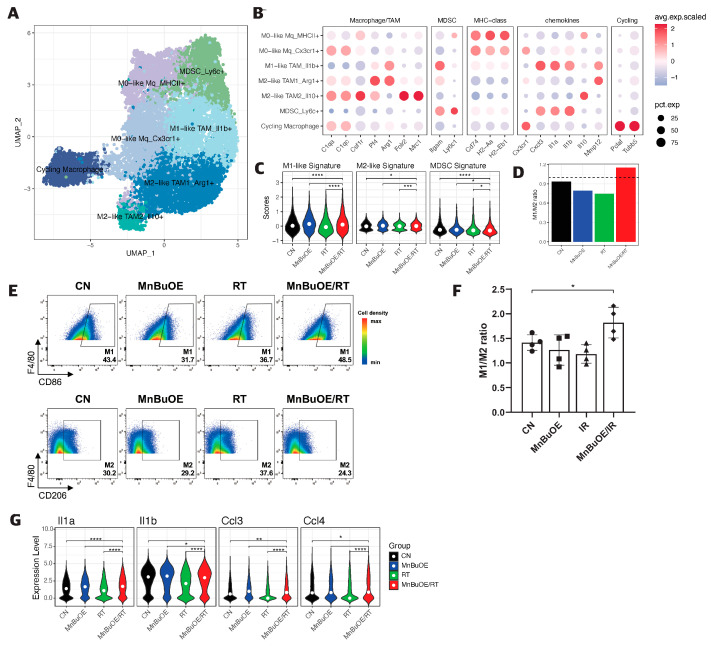
Increased pro-inflammatory macrophage proportion (M1-phenotype) in response to MnBuOE treatment coupled with irradiation. (**A**) Macrophage subtypes based on UMAP analysis presented using individual colors (n = 14,177). (**B**) Dot plot of macrophage markers for each subtype. Two clusters correspond to M0-like macrophages (MHC II+ type and Cx3cr1+ type), one cluster corresponds to M1-like TAMs (*IL1b+, Cxcl3+, Il1a+, Mmp12+*), two to M2-like TAMs (Arg1+ TAM1 type and Cd206+IL10+ TAM2 type), one to MDSCs (*Ly6C+, Itgam+, Cxcl3+, Il1a+, Il1b+*), and the last one to cycling macrophages (*Pclaf+, Tubb5+*). (**C**) Violin plot displaying the difference in M1, M2, and MDSC scores between the experimental groups. * *p* < 0.05, *** *p* < 0.001, **** *p* < 0.0001. (**D**) Ratio of M1-related macrophages (M1-like TAM_Il1b+) and M2-related macrophages (M2-like TAM1_Arg1+, M2-like TAM2_Il10+) in each experimental group. MnBuOE combined with irradiation therapy significantly augmented M1/M2 ratio of macrophages and reduced MDSCs compared with the other groups. (**E**,**F**) The flow cytometric analysis of M1- or M2-biased phenotype marker (CD86, CD206, or F4/80) for macrophage phenotype infiltrated into tumors in each experimental group. Representative density plots are shown. The ratio of M1/M2 phenotype increased in the MnBuOE/RT group compared with the other groups. * *p* < 0.05. (**G**) Differential expression of inflammatory-related genes (*Il1a, Il1b*) and M1-phenotype genes (*Ccl3, Ccl4*) among the different experimental groups. * *p* < 0.05, ** *p* < 0.01, **** *p* < 0.0001. MnBuOE, MnTnBuOE-2-PyP^5+^; UMAP, uniform manifold approximation and projection; TAM, tumor-associated macrophage; MDSC, myeloid-derived suppressor cells.

**Figure 7 antioxidants-13-00477-f007:**
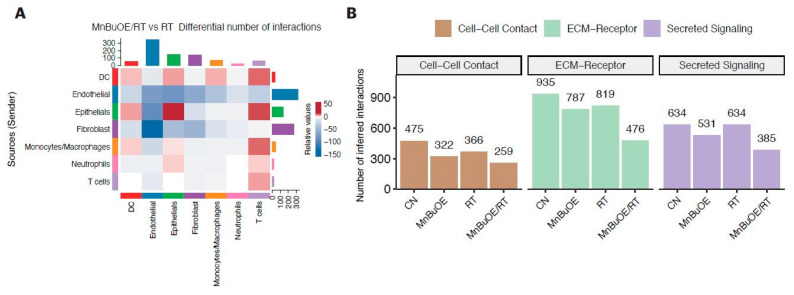
Differences in cell–cell interactions of TME following treatment with MnBuOE and irradiation. (**A**) Heatmaps showing the differential number of interactions between MnBuOE/RT (MnBuOE and irradiation) and RT (irradiation only) groups. Red color indicates increased communication in the MnBuOE/RT group compared with the RT group, whereas blue color indicates decreased possible communication. (**B**) Number of significant inferred interactions in three categories: cell–cell contact, ECM–receptor, and secreted signaling in different experimental groups. TME, tumor microenvironment; MnBuOE/RT, MnTnBuOE-2-PyP^5+^/radiation therapy; ECM, extracellular matrix.

**Figure 8 antioxidants-13-00477-f008:**
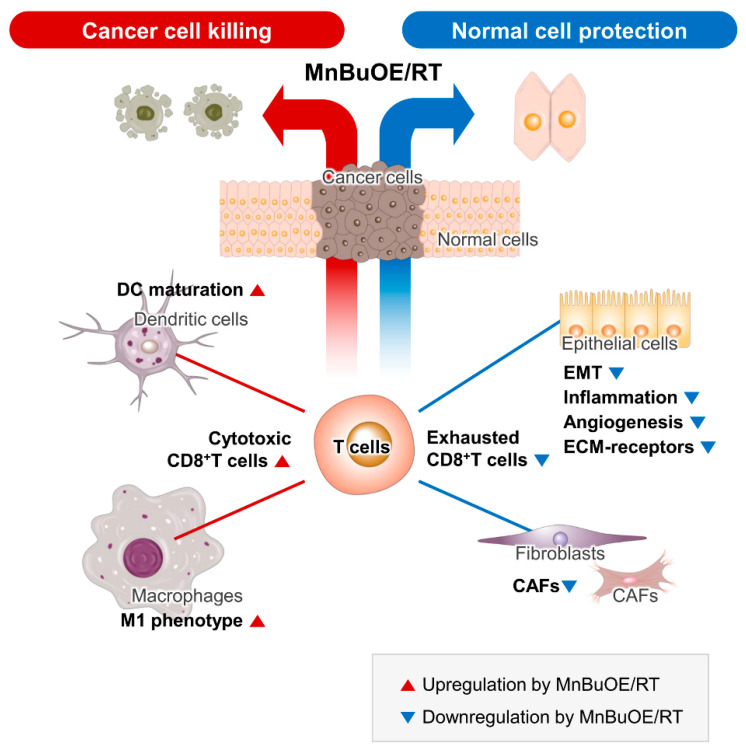
Schematic diagram of the synergistic anticancer effect of MnBuOE/RT. By analyzing the characteristics of each subtype formed by tumor and immune cells in the TME, we discussed the possibility that each cell would interact organically with other cells with cell specificity. Our data suggest that MnBuOE/RT therapy can provide a favorable environment for cancer cell removal due to the M1 macrophage, DC maturation, and augmented cytotoxic CD8 + T activity, which in turn prevents normal cell damage and metastasis, induced by inflammation and angiogenesis by inhibiting CAFs and EMT. MnBuOE/RT, MnTnBuOE-2-PyP^5+^/radiation therapy; TME, tumor microenvironment; DC, dendritic cell; CAFs, cancer-associated fibroblasts; EMT, epithelial-to-mesenchymal transition.

## Data Availability

Data are contained within the article or Appendix A.

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
