# Peer review of "Single-Cell Profiling Reveals Immune-Based Mechanisms Underlying Tumor Radiosensitization by a Novel Mn Porphyrin Clinical Candidate, MnTnBuOE-2-PyP5+ (BMX-001)"

_antioxidants, 2024, doi:10.3390/antiox13040477_

Round 1

Reviewer 1 Report

1. Seurat is used for clustering cells and annotating the cell clusters, which is the foundation of all results presented in this work. We heard too many complaints from the community about Seurat. Seurat results are very sensitive to parameter values, and researchers could get very different results by slightly changing parameters. Some researchers love this disadvantage since it gives them opportunities for fishing.

   For research integrity, I have three options to suggest: (1) redo everything with different software to conduct the task of cell annotation, such as scAnnotate a recently published paper. (2) if option one involves too much work, at least a comprehensive sensitivity analysis is needed to check whether a change in Seurat's parameter values will largely affect the results of analyzing this particular data. (3) The bottom line option is to clearly specify what parameter values are used in Seurat, acknowledge Seurat's results are not stable and discuss this as "a limitation of this work", and suggest a few better/newer methods (e.g. cite scAnnotate and scBert) for annotation (so that if the audience wants to mimic this work, they know how to do it better). 

2. Identifying subclusters of certain celltype can generate novel knowledge, but adequate validation is necessary.  I suggest authors conduct some validation work like what's done in this paper https://www.mdpi.com/2073-4409/12/24/2771 . Suppose authors do not have the resources for validation work. In that case, this must be discussed as a limitation of this work in the discussion session and provide an example of better ways to do it if you have access to the right resources (e.g. cite the reference I provided above).

Reviewer 2 Report

The authors used scRNAseq to profile cell composition changes in mice tumor after radiation and MnBuOE treatment. The rich info from the generated dataset allowed authors to investigate various cell types, which are of very use and interest to clinicians and researchers in the field. However, there are some limitations in the study that I raised below for the authors to consider and revise.

1) It will be better for the authors to make the generated dataset public available;

2) It will be better for the authors to provide animal number (n) in each group and animal number (n) pooled for scRNAseq; if the animal number was small in size, student t-test might not be proper for statistical analysis.

3) Cell death caused by radiation and MnBuOE should have been a major topic for the authors to consider, especially when monocytes/macrophages and neutrophil cells were the largest in cell composition (Fig.1), and their increased interaction with epithelial cells (Fig.7). Though the authors studied macrophages (in which monocytes were misclassified or omitted (Fig. 6)), neutrophils were omitted.

4) It will be better with context consistency for cell type annotation: macrophages should be monocytes/macrophages (Fig.1 vs. Fig.6); T cells should be NK/T cells (Fig. 1 vs. Fig.3); in figures and in the text.

5) Can authors distinguish tumor cells from normal cells, if with more tedious work on these single cells? Other than this, interpretation of data and results could be improved, so as Fig. 8. 

Other concerns:

1) algorithms/software versions were missing;

2) resolution parameters used for clustering and sub-clustering were missing;

3) references for cell type annotation were missing;

4) figure organization could be improved. for example:  Fig1. A & B. animal number in each group needed. C. NK/T cells instead of T cells. D. one gene for each cell type is appreciated. E & F replicated, not necessary. 2 panels in G could be merged into one.

5) full names for pDC and cDC are needed, so as other abbreviations in their 1st appearance.

6) sentences and grammar: for example, Lines 53-54: sentence not finished; Line 55: radio- and chemo-treatment; Line 239: "all 35,354 single cells sorted from each group" or "totally 35,354 single cells from all four groups"? please read throughout the manuscript for revision.

Round 2

Reviewer 1 Report

1. My 1st round concerns are not well addressed.

2. Authors' new results in the 4-page review response document (about consistent results got using different parameters and different methods) contradict my knowledge. As not enough details are provided to reproduce this work. I have significant concern about the correctness of the entire work.

Many authors share their code on GitHub, and if the data is new, they also submit data to GEO or other repository. This makes research reproducible. Surprising results of a non-reproducible work significantly increase our concern about correctness of the entire work.

1. scAnnotate (and many other modern supervised-learning based methods) is not dedicated to human cells. It is a supervised learning model which can handle all cells as long as it is trained with similar public data. In these days, getting such data is easy. If authors is unable to learn and use it, at least such trend need to be discussed in the paper, and literature should be properly cited.

2. I doubt authors read the paper I referred about validation. The response is not relevant to my suggestions.

Round 3

Reviewer 1 Report

My comments were reasonably addressed. 

N/A